# An evaluation of the ecological niche of Orf virus (*Poxviridae*): Challenges of distinguishing broad niches from no niches

**Rahul Raveendran Nair** [1]*, Yoshinori Nakazawa[2], A. Townsend Peterson[1]

**1** Biodiversity Institute, University of Kansas, Lawrence, Kansas, United States of America, **2** Poxvirus and Rabies Branch, Centers for Disease Control and Prevention, Atlanta, Georgia, United States of America

* rahulraveendran@ku.edu

**Data Availability Statement:** All relevant data are within the paper and its Supporting Information files.

## Abstract

Contagious ecthyma is a skin disease, caused by Orf virus, creating great economic threats to livestock farming worldwide. Zoonotic potential of this disease has gained recent attention owing to the re-emergence of disease in several parts of the world. Increased public health concern emphasizes the need for a predictive understanding of the geographic distributional potential of Orf virus. Here, we mapped the current distribution using occurrence records, and estimated the ecological niche in both geographical and environmental spaces. Twenty modeling experiments, resulting from two- and three-partition models, were performed to choose the candidate models that best represent the geographic distributional potential of Orf virus. For all of our models, it was possible to reject the null hypothesis of predictive performance no better than random expectations. However, statistical significance must be accompanied by sufficiently good predictive performance if a model is to be useful. In our case, omission of known distribution of the virus was noticed in all Maxent models, indicating inferior quality of our models. This conclusion was further confirmed by the independent final evaluation, using occurrence records sourced from the Centre for Agriculture and Bioscience International. Minimum volume ellipsoid (MVE) models indicated the broad range of environmental conditions under which Orf virus infections are found. The excluded climatic conditions from MVEs could not be considered as unsuitable owing to the broad distribution of Orf virus. These results suggest two possibilities: that the niche models fail to identify niche limits that constrain the virus, or that the virus has no detectable niche, as it can be found throughout the geographic distributions of its hosts. This potential limitation of component-based pathogen-only ENMs is discussed in detail.

## Introduction

Contagious pustular dermatitis, sore mouth, or contagious ecthyma (CE) are the well-known names of a viral skin disease that affects small ruminants around the world, particularly sheep and goats [1], with a high morbidity rate [2]. Contagious ecthyma is caused by *Orf virus*, an epitheliotropic parapoxvirus of the family *Poxviridae* [3]. Initial rise in body temperature [4]

**Funding:** The author(s) received no specific funding for this work.

**Competing interests:** The authors have declared that no competing interests exist.

and subsequent development of lesions on both cornified epithelium and mucosal surfaces [5] are the characteristic clinical presentations of CE in ruminants. CE lesions are commonly seen in the regions of orofacial surfaces, udder, teat, and genital organs in animals [1, 4, 6–8]. Orf virus infection is of significant public health concern owing to frequent transmission of disease from animals to humans [9–11]. In humans, systemic effects such as fever, malaise, and lymph node inflammation [12] are most often accompanied by development of inflamed papules on the dorsal aspect of the hands [5]. Infection occurs less often on the face [13, 14], nose [15], armpit [16], scalp [17], genitals [18], perineum [19], and eyes [20]. Given their self-limiting nature [4], CE infections have not seen much research attention over the years [21].

Livestock farming, an important source of income for people living in developing countries [21], has seen significant financial losses owing to Orf virus infections [22, 23]. Negative impacts on financial viability are mainly in the form of reduced weight gain, requirement of additional feed [20, 24], decreased quantity and quality of milk production, and occasional death of infected ruminants [25]. Orf virus can remain viable in certain environments [4] and on animal skin and fomites for months or years [5, 26]. Loss of epithelial barrier intactness (*i.e.*, skin abrasions) is one major factor that contributes to Orf virus infection in animals [5, 20, 27, 28]. Grazing is one of the major routes of viral transmission from the environment to host animal through the skin scrapes on the orofacial regions as a result of the consumption of dried feeds [4]. Animal-to-animal and animal-to-human transmission of virus occurs through daily or accidental close contacts with infected animals or contaminated fomites [4, 21, 29, 30]. Human-to-human transmission of Orf virus rarely occurs [13]. In view of its transnational spreading capacity and human health concerns, Orf virus-associated infections demand careful examination [11, 21, 31].

Emerging and re-emerging viral diseases are of major public health concern in the 21st century [32], a fact underlined by the recent COVID-19 pandemic. Major causative factors that contribute to the emergence or re-emergence of infectious diseases can be classified into three: (1) alterations in human-animal interactions, (2) modifications of the environment or ecosystem, and (3) mutation and evolution of pathogens [33]. In a distributional ecology framework, focus is on environmental conditions and their influence on pathogen circulation and transmission as a means by which to characterize and map potential distributional areas of pathogens in geographic space [34].

In this study, we attempted to develop correlative models of the climatic niche spaces of Orf virus worldwide to gain insight into possible transmission patterns for disease risk management. Component-based, pathogen-only ecological niche models (ENMs) [34, 35] based on pathogen occurrence records and abiotic environmental variables were employed in this study. Given the broad occurrence of the virus, we focus on the initial question of whether niche limits can be discerned and documented *versus* whether the virus is able to persist and infect animals in most parts of the world under any set of conditions.

## Materials and methods

### Input data

Occurrence records of Orf virus (*i.e.*, incidences of Orf infections; Fig 1) were collected from published literature in the PubMed database and Google Scholar, using the search terms "Orf", "Orf virus", "Orf virus outbreak", "contagious ecthyma", "CE virus", "ovine pustular dermatitis", "contagious pustular dermatitis", and "CE outbreak". The search yielded 229 presence points for the period 1971–2021 (S1 Table). We obtained only a single record from the Global Biodiversity Information Facility (GBIF; http://www.gbif.org, DOI: https://doi.org/10.15468/dl.dzhufb). As suggested by Peterson and Samy [36], other than removing duplicate

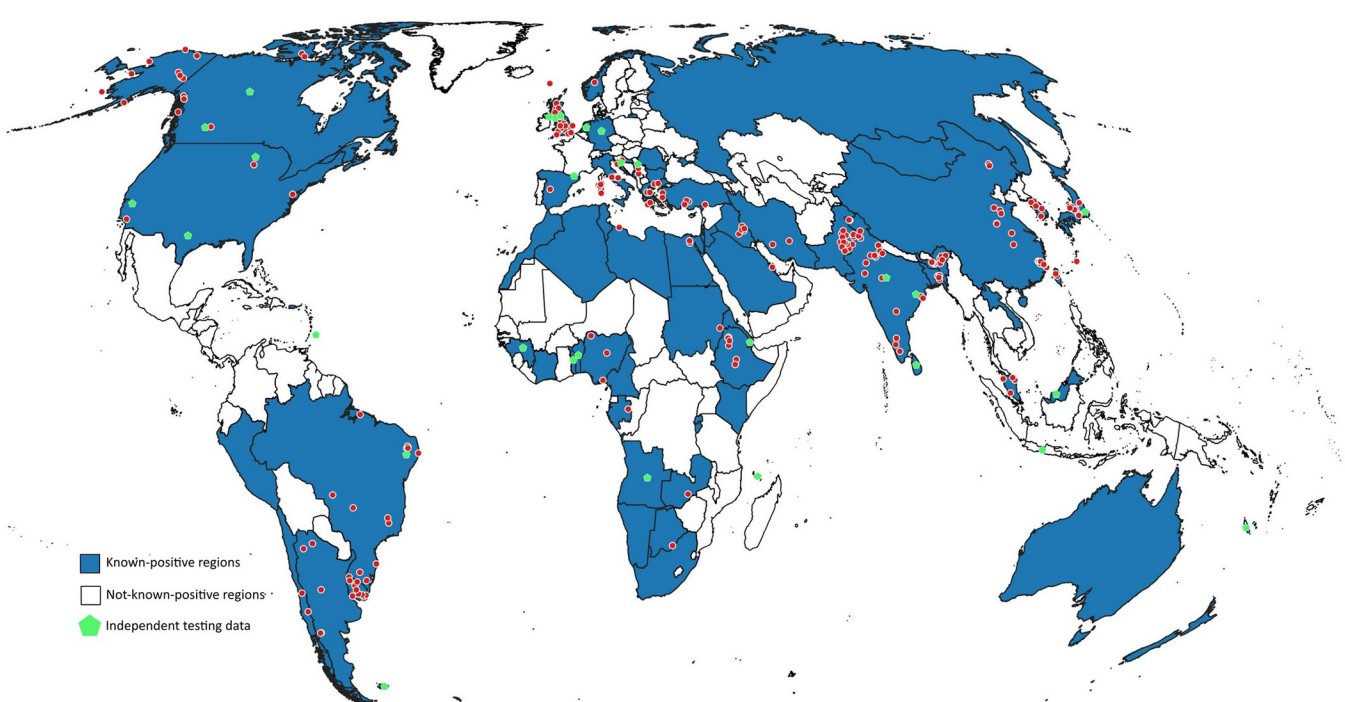

**Fig 1. Known global distribution of Orf virus.** Red circles indicate the occurrence points used in the present study; Green pentagons indicate the position of independent testing data points. Free vector data were sourced from Natural Earth (naturalearthdata.com).

occurrence records, we chose not to take steps to reduce spatial autocorrelation in the dataset prior to modeling because higher data density in certain regions may (or may not) reflect a biological phenomenon; in spatial epidemiology studies, it is reasonable to capture such signals in modeling efforts [34].

To account for uncertainty associated with presence points in linking occurrences to corresponding environmental characteristics, circular buffer zones were created around each record, with a radius depending on the amount of uncertainty associated with each point. For occurrences with geo-coordinate information (*i.e.*, relatively precise occurrence records), a buffer zone of radius 5 km was assigned. For occurrences with no geocoordinate information, a four-step process was adopted: identification of the finest administrative division corresponding to the locality (*i.e.*, town, municipality, district or state), determination of the centroid of that administrative area, calculation of its geographic extent, and estimating the radius of the calculated area. This latter step assumed that locality is circular in shape ($r = \sqrt{\frac{A}{\pi}}$, where A = calculated area) (Fig 2 and S2 Table). This step has the advantage of casting the random-representative points within a consistent area surrounding the centroids of administrative areas, but the disadvantage of leaving out the irregular, more-distant corners of those same areas; more detailed testing and exploration may reveal more about the relative merits of this step. Areas of localities were calculated using QGIS desktop version 3.18.3 Zurich [37] and Google Maps. To incorporate the uncertainty directly in the development of our models (see references [36, 38]), we generated five random points within each circular buffer zone, to form five replicate occurrence datasets (R1-R5), to be used as input occurrence data for building models (S3 Table).

The area '**M**' in the BAM framework is the geographic area to which a species has had access over relevant time scales [39], and represents a key element in model calibration [39–41].

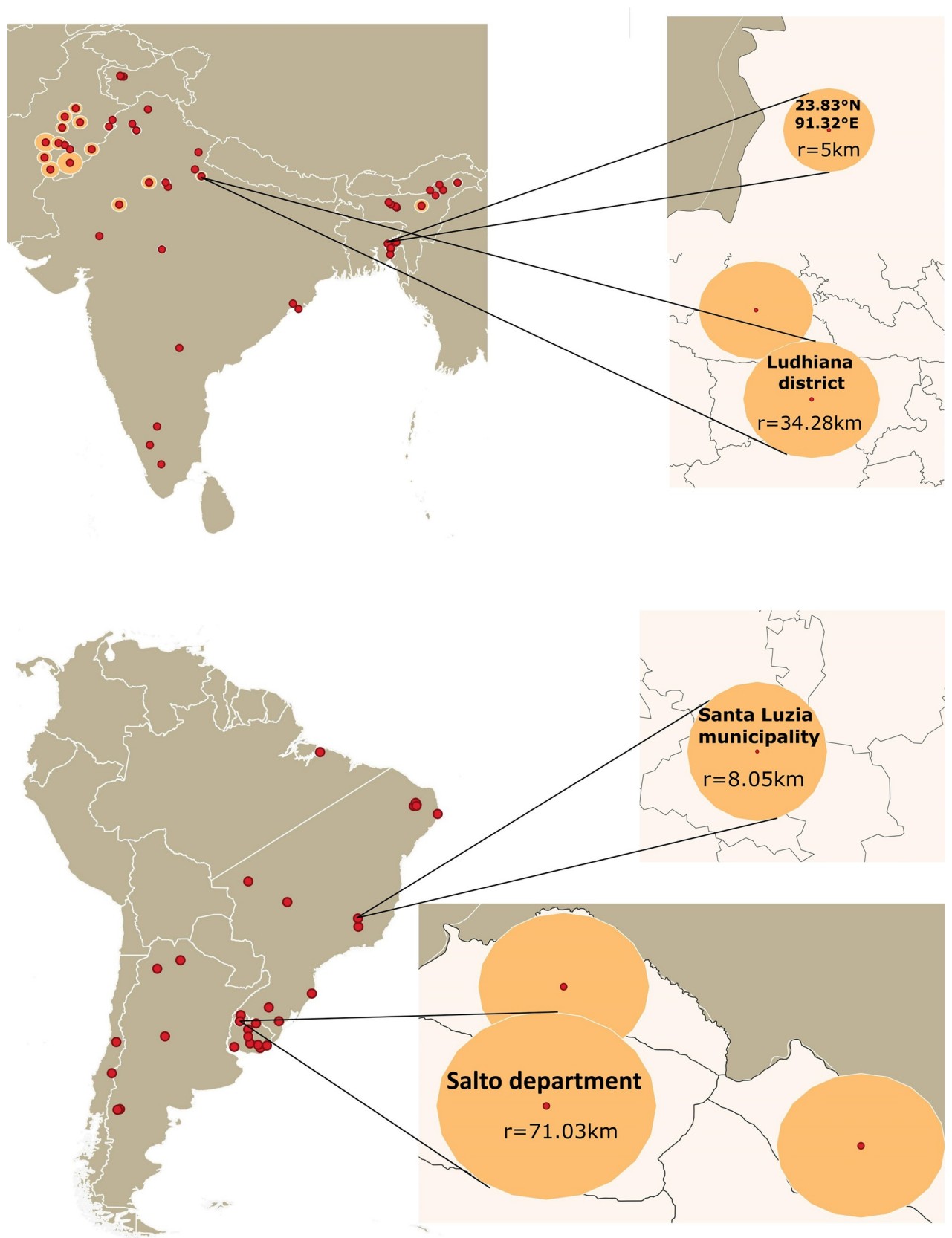

**Fig 2. Uncertainty allocation.** Representative buffer zones of varying radius representing uncertainty in geolocalization based on the area of different localities. Free vector data were sourced from Natural Earth (naturalearthdata.com).

Considering the global distribution of Orf virus [42], we included the entire land area of the Earth, excluding Antarctica, as a hypothesis of **M** and as a model calibration area.

Climatic data layers (19 bioclimatic variables at 10' spatial resolution or ~17 km) were downloaded from the WorldClim version 2 database [43]. Although a temporal mismatch exists between the climatic data layers (averaged over the period 1970–2000) and the reporting time frames of Orf cases in our dataset (1971–2021), we retained the occurrence records from after 2000 for two reasons: long-term and persistent presence of Orf virus in the environment [4, 23, 44], and high resilience of Orf virus to environmental degradation [45]. Four layers (mean temperature of wettest quarter, mean temperature of driest quarter, precipitation of warmest quarter, precipitation of coldest quarter) were removed owing to known spatial artifacts associated with those data layers [46]. The environmental variables selected are highly correlated and/or non-independent [47], and multi-collinearity among these predictor variables can induce errors in model training process [48], resulting in dubious assumptions [49–51]. We performed principal component analysis (PCA) on the remaining 15 variables to assure orthogonality and minimize the dimensionality of models (S4 Table) [46] using kuenm_rpca function of kuenm R package [52]. Occurrence data partitioning exercises were performed using caTools R package version 1.18 [53].

## Data processing

As part of the selection process of model parameters, an initial modeling experiment was performed by randomly splitting the occurrence points of one replicate set into training (70%) and testing (30%). All 29 combinations of five feature classes (linear—l, quadratic—q, hinge—h, product—p, threshold—t) and 12 regularization multiplier values (0.1, 0.3, 0.5, 0.75, 1.0, 1.5, 2.0, 2.5, 3.0, 5.0, 7.0, 10.0) were used in combination with five principal components (PCs: 1–5). Based on the results of this initial modeling, we extended the number of PCs under testing using 10 different environmental datasets (PCs: 1, 1–2, 1–3, 1–4, 1–5, 1–6, 1–7, 1–8, 1–9, 1–10), and performed a second modeling experiment without changing combinations of feature class and regularization multiplier values.

Upon assessing the models developed from the first and second experiments, we evaluated 20 models calibrated using different random partitions of the five replicate occurrence sets. This computationally expensive modeling step was developed specifically to take into account the effects of positional uncertainty on model outcomes, thus not overinterpreting models based on points not placed with complete precision. A replicate dataset was subjected to two ways of partitioning (three-partition sets and two-partition sets) (for details refer to Fig 3). A completely independent test data set of 29 occurrence records was sourced from the online portal of the Centre for Agriculture and Bioscience International (CABI; https://www.cabi.org/isc/datasheet/88087#tohostAnimals, accessed on 4 July 2022). In the end, 29 combinations of feature classes, eight regularization multiplier values (0.1, 0.3, 0.5, 0.75, 1, 1.5, 2, 2.5) and six sets of climatic variables (PCs: 1–5, 1–6, 1–7, 1–8, 1–9, 1–10; note that simpler environmental datasets and higher regularization multiplier values were not considered further based on the results of the earlier experiments) were employed in the final modeling experiment. Final models were selected from among candidate models based on three criteria: statistical significance, omission rate, and lowest Akaike information criterion (AICc) values, as suggested by Cobos et al. [52].

**Three-partition experiment**

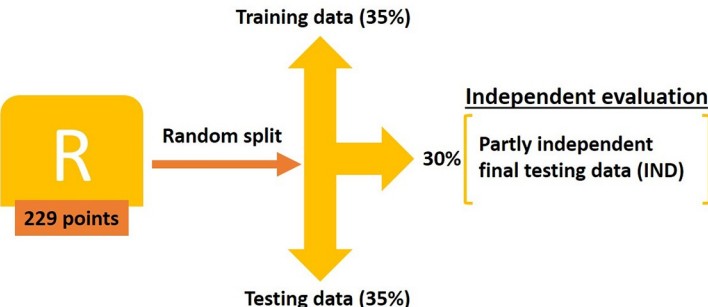

**Two-partition experiment**

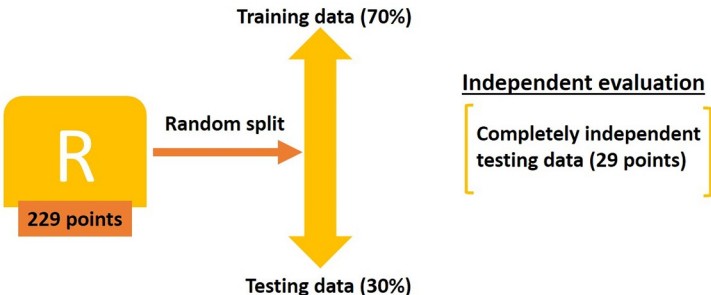

**Fig 3. Scheme of partitioning the replicate datasets (R) for creating two-partition and three-partition sets.** Each replicate dataset was subjected to random partitioning two times.

Statistical significance of models was tested using partial receiver operating characteristic (*p*ROC) statistics [54]. Statistically significant models were filtered using omission rate, retaining only those models for which omission rate was <5% [55]. Statistically significant models with <5% omission rate were ranked based on AICc values and the models that are within 2 AICc units of the minimum were chosen as final models [56]. In traditional AUC approach, commission and omission errors are given equal weight in assessing the model performance [57]. As we used only presence-only data, partial AUC ratio, determined using the partial ROC method, was employed to analyze the model performance [54]. In this method, the x-axis of traditional ROC AUC (commission error) is replaced with proportion of the whole study area predicted as present to restrict AUC calculations to regions where the model predicts [54]; under the restricted curve, only the relevant regions that meet the user-defined omission threshold ($E$ = 5%) were considered. Partial AUC ratio is defined as the ratio of AUC of the restricted ROC curve to the AUC of the restricted random expectation curve [54]. Partial AUC ratio values range from 0 to 2; a value of 1 indicates random performance [54, 58]. Partial AUC ratios were averaged across 10 replicates of final models, and represented as mean partial AUC ratios. From the set of final models, the best model was identified as the one with a delta_AICc value of zero (*i.e.*, difference in AIC value between the top model and the remaining models in the final set). The kuenm_ceval function of kuenm R package [52] was used for calculating statistical significance, omission rate and AICc values. Model calibration and evaluation experiments were performed using the kuenm R package [52]. Uncertainty among the model predictions was estimated as the range of suitability values, generated as the difference between maximum and minimum values among 10 model replicates [59].

## Niche visualization in environmental space

Typical ENM outputs display the footprint of an ecological niche model in geographic space. In reality, however, ecological niches of species are manifested in environmental space; as such, visualization of niches in environmental space in tandem with maps in geographic space allows better understanding and interpretation of models [60]. As fundamental niches are assumed to be convex in shape [61–63], they can be approximated using minimum volume ellipsoids (MVE) [64]. As such, to complement the Maxent-based niche models, we estimated the niche of Orf virus as an MVE in environmental space using NicheA [60]. Seventy percent of replicate occurrence data and first three PCs (PC1, PC2, and PC3) were used to implement the MVE. Moment-based minimum-volume ellipsoids [65] were used to implement MVE separately to visualize training points and 29 independent testing points. A brief comparison of different tools for these analyses is offered in the form of a table in the Supplementary Materials of this contribution (S8 Table).

## Results

The first pass of our modeling experiments for model complexity optimization yielded six candidate models meeting the selection criteria (S5 Table) out of 60 candidate models. They consisted of simple models with linear features only, and relatively low regularization values (<2.0). Statistically significant partial AUC ratios (mean ratio 1.116 ± 0.002, all $P < 0.001$) suggested that the predictions of those models were better than random predictions. The second modeling experiment also selected six candidate models (S6 Table), in this case out of 600 models, with a slight increase of complexity compared to the previous models: models had both linear and quadratic response types. Larger sets of predicted variables (set 10: 1–10 PCs, set 9: 1–9 PCs) were preferred in the selected models. These models generally had lower regularization values (0.3, 0.5, 0.75).

Based on the two rounds of initial explorations of modeling experiments, a subset of lower regularization values (0.1, 0.3, 0.5, 0.75, 1, 1.5, 2, 2.5) and six relatively rich environmental datasets [set 5 (1–5)–set 10 (1–10)] was explored in combination with all 29 combinations of feature classes in the context of the 20 replicate occurrence datasets detailed above. In three-partition experiments, it was interesting to note that model predictions evaluated using the conventional strategy (*i.e.*, testing predicted model with test data drawn as a random subset of the pool of occurrence data) reported that 8 of 10 (80%) models met our omission rate criteria (<5%) (Table 1). Two-partition experiments also had 80% of selected models meeting omission rate criteria (Table 2). Here, a consistent and repeated pattern of model success in terms of inductive reasoning (generalizations drawing from specific observations based on evaluation metrics) provided an optimistic impression about model quality. Information pertaining to total numbers of models assessed and selected in both two- and three- partition experiments are provided in S7 Table.

However, manual assessment of models (Fig 4) based on the available distributional knowledge of Orf virus revealed that all of our models failed consistently to predict known occurrences in many parts of the world (e.g., Russia, Brazil, Africa, Argentina, western North America, Alaska, Canada, and China; see discussion below). We noted that models developed using linear feature class alone were able to predict potential occurrences in these regions relatively better (Fig 5). Evaluating the selected models using two partly independent test datasets (IND1 and IND2) sourced from the occurrence pool (*i.e.*, three-partition models) revealed that only one model out of 10 final models met our omission rate criteria (<5%). No model met our omission rate criteria among the 10 models based on two-partition sets. All final models in both two- and three- partition experiments were statistically significant (P < 0.001,

**Table 1. Model evaluation summary of three-partition experiments.**

| Replicate | Set | Best models selected | Mean partial AUC ratio | OR | AICc | Independent final evaluation | |
|---|---|---|---|---|---|---|---|
| | | | | | | IND1 | IND2 |
| | | | | | | OR | |
| R1 | CAL_1 | M_2.5_F_lq_Set_5 | 1.130 | 0.089 | 3806.141 | 0.043 | 0.044 |
| | CAL_2 | M_1_F_l_Set_6 | 1.134 | 0.038 | 3911.748 | 0.101 | 0.102 |
| R2 | CAL_1 | M_0.1_F_lqp_Set_7 | 1.322 | 0.050 | 3811.500 | 0.164 | 0.088 |
| | CAL_2 | M_0.1_F_l_Set_6 | 1.150 | 0.038 | 3959.329 | 0.059 | 0.088 |
| R3 | CAL_1 | M_0.1_F_lqp_Set_6 | 1.345 | 0.026 | 3773.834 | 0.151 | 0.104 |
| | CAL_2 | M_1.5_F_lqpt_Set_10 | 1.299 | 0.039 | 3729.644 | 0.090 | 0.029 |
| R4 | CAL_1 | M_0.1_F_lqp_Set_6 | 1.358 | 0.038 | 3797.843 | 0.147 | 0.088 |
| | CAL_2 | M_0.5_F_l_Set_6 | 1.133 | 0.038 | 3962.877 | 0.058 | 0.088 |
| R5 | CAL_1 | M_0.1_F_lqp_Set_7 | 1.335 | 0.038 | 3769.861 | 0.164 | 0.088 |
| | CAL_2 | M_0.5_F_l_Set_6 | 1.128 | 0.039 | 3907.445 | 0.059 | 0.088 |

OR = Omission rate. CAL_1 and CAL_2 represent two distinct random subsets drawn from available occurrence data. Name of models provides details pertaining to the regularization multiplier value, feature class and environmental dataset (e.g. M_2.5_F_lq_Set_5 indicates regularization multiplier value: 2.5, feature classes: linear and quadratic, Set_5: climatic dataset containing five PCs). IND1 and IND2 represent the partly independent testing datasets.

Tables 1 and 2). Results of independent final evaluation of selected models thus coincided with manual assessment of predicted models based on distributional knowledge (see Discussion for more details): our models failed to predict distributional potential of Orf virus in several areas across the world. Uncertainty estimations revealed a general but non-uniform trend that high predicted habitat suitability was often associated with medium to high uncertainty (S1 Fig.).

The occurrence points used to calibrate Maxent models were displayed in environmental space (Fig 6A) in relation to environmental combinations in existence around the world (Fig 6B). Similarly, the environmental space falling outside the MVE summarizing occurrences of the virus was represented in geographic space (Fig 6C). Areas falling outside of the Orf MVE were distinct and cohesive, and scattered around the world. Climatic conditions of these

**Table 2. Model evaluation summary of two-partition experiments.**

| Replicate | Set | Best models selected | Mean partial AUC ratio | OR | AICc | Independent final evaluation |
|---|---|---|---|---|---|---|
| | | | | | | 29 points |
| | | | | | | OR |
| R1 | CAL_1 | M_0.1_F_lqp_Set_10 | 1.316 | 0.044 | 5375.808 | 0.241 |
| | CAL_2 | M_2_F_lqpt_Set_8 | 1.263 | 0.047 | 5399.658 | 0.172 |
| R2 | CAL_1 | M_0.1_F_l_Set_6 | 1.077 | 0.059 | 5696.234 | 0.068 |
| | CAL_2 | M_1_F_lqp_Set_9 | 1.289 | 0.045 | 5451.725 | 0.207 |
| R3 | CAL_1 | M_0.1_F_l_Set_6 | 1.079 | 0.061 | 5619.882 | 0.068 |
| | CAL_2 | M_0.1_F_lqp_Set_8 | 1.336 | 0.045 | 5372.768 | 0.172 |
| R4 | CAL_1 | M_0.1_F_lqp_Set_8 | 1.342 | 0.043 | 5455.684 | 0.172 |
| | CAL_2 | M_2_F_lqpt_Set_9 | 1.385 | 0.044 | 5427.347 | 0.276 |
| R5 | CAL_1 | M_0.3_F_lqp_Set_8 | 1.236 | 0.046 | 5403.969 | 0.138 |
| | CAL_2 | M_2_F_lqpt_Set_10 | 1.351 | 0.029 | 5383.063 | 0.207 |

OR = Omission rate. CAL_1 and CAL_2 represent two distinct random subsets drawn from available occurrence data. Name of models provides details pertaining to the regularization multiplier value, feature class and environmental dataset (e.g. M_0.1_F_lqp_Set_10 indicates regularization multiplier value: 0.1, feature classes: linear (l), quadratic(q), and product(p), Set_10: climatic dataset containing 10 PCs).

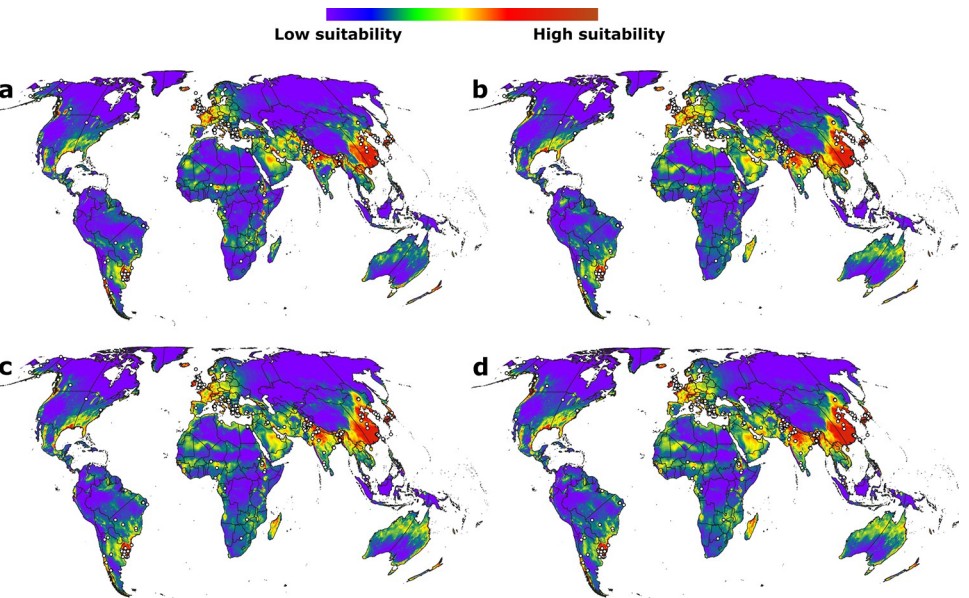

**Fig 4. Representative models.** a-d) Maxent models displaying consistent and repeated omission of known distributional areas of Orf virus. White circles indicate the known occurrence records. Free vector data were sourced from Natural Earth (naturalearthdata.com).

omitted geographic regions (e.g., Russia) in the Maxent models appeared to be within the MVE, and yet climatic conditions excluded from the MVE cannot be considered as unsuitable owing to the known global distribution of Orf virus. As such, the ability of either class of models that we developed (Maxent and MVE) to discriminate between suitable and unsuitable conditions for Orf virus appears to be dubious.

## Discussion

Our study focused primarily on the ability of correlative models to predict suitability of regions for Orf virus across the world; this prediction-focused modeling focus contrasts with explanatory models that would attempt to explain the importance of different environmental variables. Principal components analysis transforms highly correlated environmental variables into a smaller set of orthogonal (*i.e.*, non-correlated) axes, retaining most of the original information in the original, raw (untransformed) environmental data [48, 67], summarizing environmental variability across a given geographic area [49, 68]. In our analyses, multi-collinearity reduction via PCA identified 10 axes that together explained >95% of the variation in the original 15 environmental variables [48, 68].

In spatial epidemiology, the lack of knowledge regarding the precise location of outbreaks and possibilities of hosts, moving away from the exact location prior to the initial documentation of infection can affect the model quality by reducing the effective sample size [38]. Hence, in this study, traditional single-point-based representation was replaced with circular buffer zones, containing random points to account for the level of uncertainty associated with each outbreak [36]. This method resembles a spatial bootstrap that adds more weights to occurrences with known geocoordinate information than occurrences without such details, and thus enhances our ability to deal with varying degrees of uncertainty associated with precise localization [38].

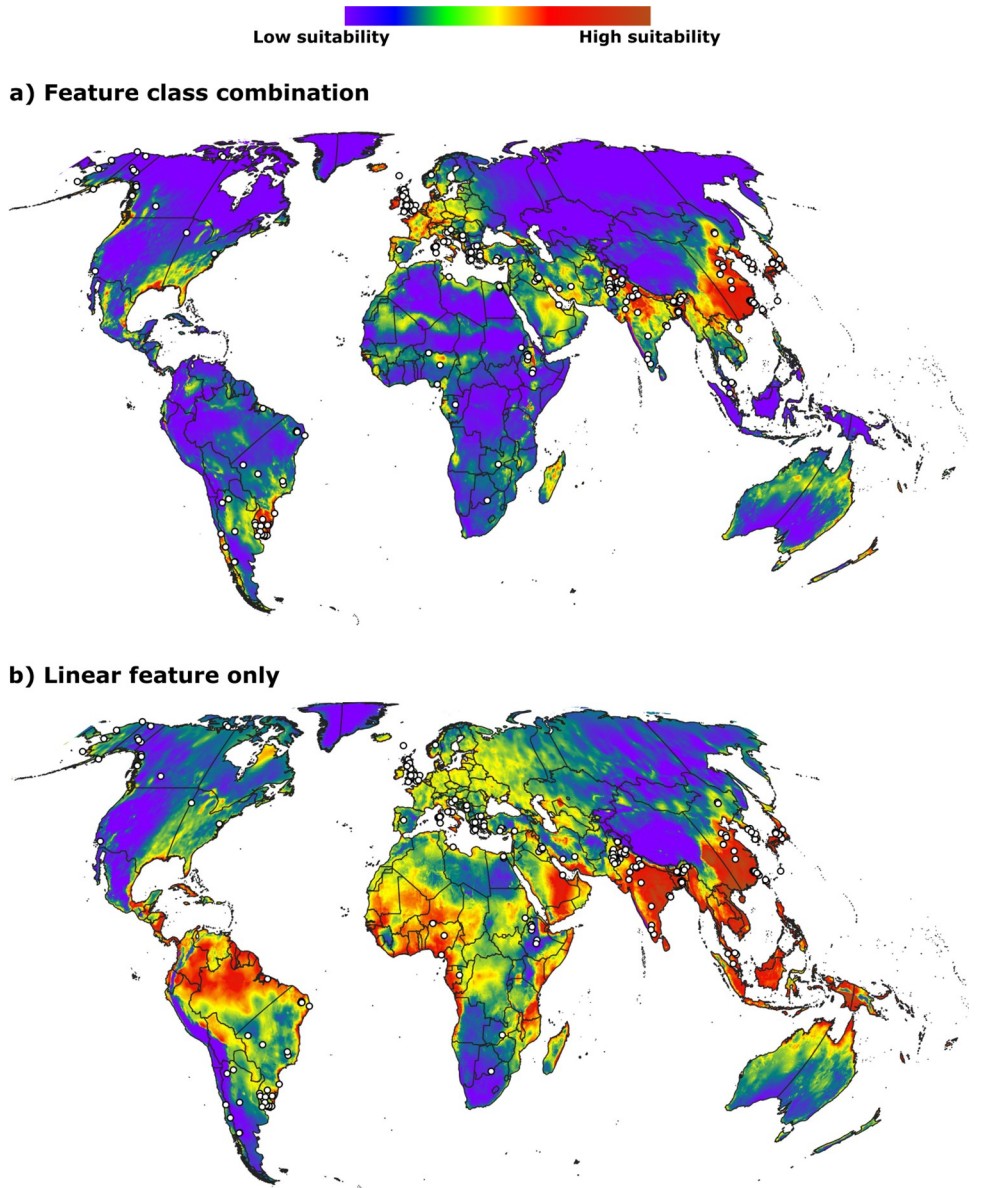

**Fig 5. Difference in prediction.** a) Model that integrated multiple feature classes, and b) linear model displaying relatively better prediction of Orf prevalence in known distributional areas. Feature classes represent different ways of transforming original environmental variables (linear = no transformation of variables, quadratic = square of variables, product = pairwise multiplication of variables, threshold = transforming variables using a discrete stepwise function, and hinge = transforming variables using a smoothed stepwise function) [66]. Free vector data were sourced from Natural Earth (naturalearthdata.com).

A fundamental question in disease biogeography is whether pathogens have ecological niches of their own, or whether their occurrences simply reflect the ecological niches of their hosts [69, 70]. In this study, we found models that appeared to be significantly better than random in their predictions. We sought to answer the niche question for Orf virus via two assessments: (1) Do generated models show non-random patterns of model failure? (2) Can we see consistent and repeated patterns of omission in the models generated? In this latter step, we assessed the models developed by integrating additional distributional knowledge, and noticed

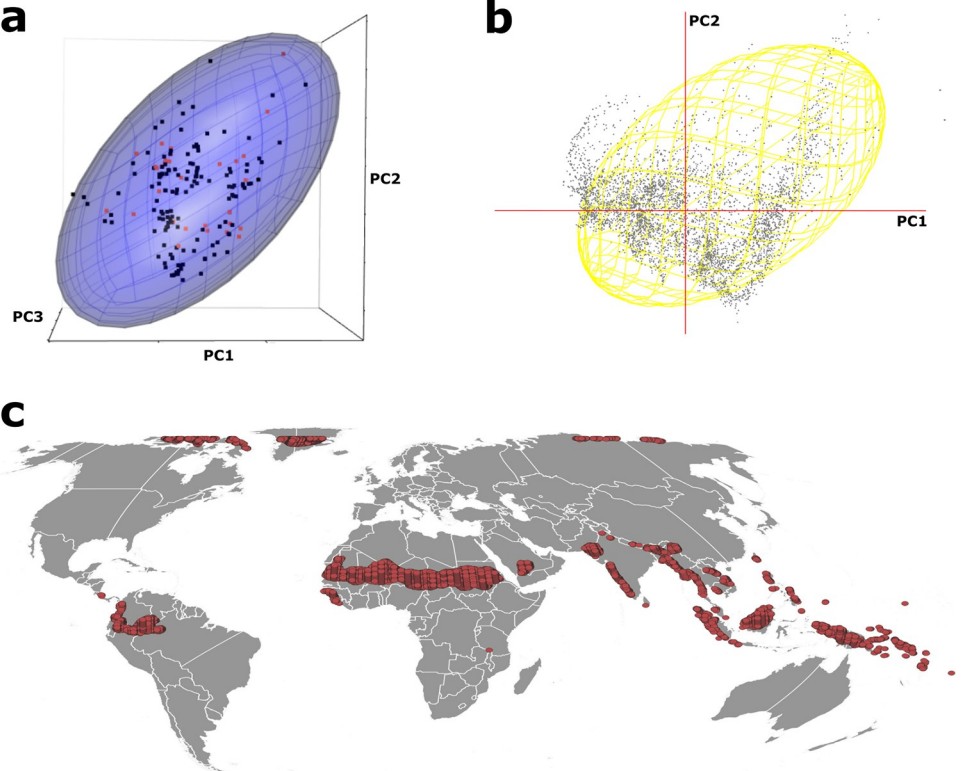

**Fig 6. MVE model and visualization of regions left out of niche estimates.** a) Occurrence points used to calibrate Maxent models are displayed in MVE using first three PCs. The dark blue spots indicate the training points, and red spots indicate the independent test data, b) two-dimensional environmental background cloud generated using first two PCs, and c) environment outside MVE is plotted in geographic space (G) to identify unsuitable niches. Free vector data were sourced from Natural Earth (naturalearthdata.com).

that significant parts of the known distribution of Orf virus were not represented in the geographic space visualizations.

Failure of models to predict known distributional areas is reflected in nine areas, scattered across North America, South America, Europe, Africa, Asia, and Australia. For example, during 1935–1992, potential for Orf virus infection was investigated and developed as an agroterrorism agent in Russia (former USSR) [71, 72], and several reports discussed the Orf infection of small ruminants in Russia [73–77]. Considering these concrete (but non-specific) records of the virus within Russia, models were expected to predict distributional potential there, even though no presence points were available to us from Russia. The absence of predicted potential for the virus in Russia indicates that the models are leaving out known distributional areas.

Multiple cases of Orf infections have been documented in Alaska [78–80] yet none of the models predicted distributional potential in Alaska. Similarly, for Canada, except in the southwestern coastal regions and a small region in Alberta by a few models, distributional potential was not predicted in any other part of the country in spite of the documented occurrence of Orf virus in Canada [81–84]. Numerous scientific articles have reported Orf cases from various parts of China [85–92] yet predicted distributional potential was restricted to parts of central and southeastern China.

Other regions of likely model failure were in south-central Argentina [93–95], western North America [96–98], Brazil [99–103], Africa [104–111] and Australia [112–115]. These observations of consistent and repeated patterns of omission demonstrate that our models are

of inferior quality in predicting Orf virus "niche" because model quality is a function of the model's predictive ability in identifying distributional areas of the species in question [55, 116].

Climatic niches are manifested in environmental space [60]. Visualization of species' niches in environmental space offered a complementary approach to checking possible biases in representation of species' niche in geographic space [60, 117–119]. Exploration and visualization of climatic conditions left out of the MVE models in relation to known distributional areas for Orf virus in environmental space further confirmed the inferior quality of our models, as "left-out" areas in several cases are known to hold the virus (see above summary).

These results underline the need for validating models using data completely independent of the occurrence points used in model calibration [120, 121]. Using testing data derived via random partitioning of the same dataset may not be independent of the training data [121], in view of increased data density of Orf virus distribution in certain geographic regions, and as can be appreciated in Fig 1. Final evaluation using these ostensibly independent test data revealed a realistic inference on model quality: 19 out of 20 models failed the final independent evaluation, in spite of the "independent" test data not being spatially separate from the training data.

From the statistical point of view, the models that we developed, although they made predictions that were significantly better than random predictions, proved to be of inferior quality (*i.e.*, they had high omission rates) when subjected to independent final evaluation. Considering the fact that obligate pathogens can be found across the whole geographic distributions of known hosts, this may be a case in which the pathogen does not manifest a detectable ecological niche separate from that of its host [35, 69]. Distribution of pathogens strictly as a function of host availability has already been reported in a few studies such as tuberculosis in cattle [122], intersecting ranges of *Phlebotomus papatasi* (vector) and *Leishmania major* (pathogen) [123], rabies virus in vampire bats causing infection to livestock [124], and dengue virus compared with its vector *Aedes aegypti* [35]. However, confirming whether or not Orf virus has a detectable niche separate from that of its host remains a challenge, awaiting much larger sample sizes, and information on host-breadth [35] (*i.e.*, accessible host populations for Orf virus to infect and survive in the absence of dispersal barriers). Host-breadth identification may reveal relevant biotic factors that can inform component-based ENM of pathogens [35].

## Supporting information

**S1 Fig. Uncertainty representation.** a-d) Median output of Maxent models displaying habitat suitability predictions. White circles indicate the known occurrence records, and e-h) corresponding uncertainties associated with those model predictions estimated as the range of suitability values among 10 model replicates. Free vector data were sourced from Natural Earth (naturalearthdata.com).
(TIF)

**S1 Table. Presence points.** Details of occurrence records of Orf virus.
(DOCX)

**S2 Table. Uncertainty estimation.** Assigning uncertainty based on the geographic extension of administrative area.
(XLSX)

**S3 Table. Replicate datasets.** Five replicate occurrence datasets used as input occurrence points for building models.
(XLSX)

**S4 Table. Principal component analysis.** Loadings along the PCs and proportion of variance explained by each of the PCs.
(XLSX)

**S5 Table. Model evaluation I.** Performance of models generated in the first pass of modeling experiment.
(XLSX)

**S6 Table. Model evaluation II.** Performance of models generated in the second modeling experiment.
(XLSX)

**S7 Table. Assessment and selection.** Total number of models assessed and selected in both two- and three- partition experiments.
(XLSX)

**S8 Table. Modeling platforms: Major features of Maxent and NicheA modeling platforms.**
(DOCX)

## Acknowledgments

The authors thank KU ENM group and the Biodiversity Institute, University of Kansas for providing facilities for this work.

   **Disclaimer:** The findings and conclusions in this report are those of the author(s) and do not necessarily represent the views of the Centers for Disease Control and Prevention.

## Author Contributions

**Conceptualization:** Rahul Raveendran Nair, Yoshinori Nakazawa, A. Townsend Peterson.

**Data curation:** Rahul Raveendran Nair, A. Townsend Peterson.

**Formal analysis:** Rahul Raveendran Nair.

**Investigation:** Rahul Raveendran Nair, Yoshinori Nakazawa.

**Methodology:** Rahul Raveendran Nair, Yoshinori Nakazawa, A. Townsend Peterson.

**Project administration:** Rahul Raveendran Nair, A. Townsend Peterson.

**Resources:** Yoshinori Nakazawa, A. Townsend Peterson.

**Software:** Rahul Raveendran Nair, A. Townsend Peterson.

**Supervision:** Yoshinori Nakazawa, A. Townsend Peterson.

**Validation:** Rahul Raveendran Nair, Yoshinori Nakazawa, A. Townsend Peterson.

**Visualization:** Rahul Raveendran Nair, Yoshinori Nakazawa, A. Townsend Peterson.

**Writing – original draft:** Rahul Raveendran Nair.

**Writing – review & editing:** Rahul Raveendran Nair, Yoshinori Nakazawa, A. Townsend Peterson.

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
