## [Decision Letter · Decision Letter 0]

6 Jul 2023

PONE-D-23-15637An Evaluation of the Ecological Niche of Orf Virus (Poxviridae): Challenges of Distinguishing Broad Niches from No NichesPLOS ONE

  Dear Dr. Nair,

Thank you for submitting your manuscript to PLOS ONE. After careful consideration, we feel that it has merit but does not fully meet PLOS ONE’s publication criteria as it currently stands. Therefore, we invite you to submit a revised version of the manuscript that addresses the points raised during the review process.

 Both reviewers made important remarks and I'd ask you to address all of them as you manuscript, if accepted after a major review, might bring some important contributions to the field.

We look forward to receiving your revised manuscript.

Kind regards,

Paulo Brandao

Academic Editor

PLOS ONE

“The first author would like to thank Mr. Ajayya Kumar (COO, Emircom, Abu Dhabi) for funding the research stay at the University of Kansas. The authors thank KU ENM group and the Biodiversity Institute, University of Kansas for providing facilities for this work.”

5. We note that [Figure(s) #] in your submission contain [map/satellite] images which may be copyrighted. All PLOS content is published under the Creative Commons Attribution License (CC BY 4.0), which means that the manuscript, images, and Supporting Information files will be freely available online, and any third party is permitted to access, download, copy, distribute, and use these materials in any way, even commercially, with proper attribution. For these reasons, we cannot publish previously copyrighted maps or satellite images created using proprietary data, such as Google software (Google Maps, Street View, and Earth). For more information, see our copyright guidelines: http://journals.plos.org/plosone/s/licenses-and-copyright.

1. You may seek permission from the original copyright holder of Figure(s) [#] to publish the content specifically under the CC BY 4.0 license. 

Reviewers' comments:

Reviewer's Responses to Questions

**Comments to the Author**

1. Is the manuscript technically sound, and do the data support the conclusions?

Reviewer #1: Partly

Reviewer #2: Partly

2. Has the statistical analysis been performed appropriately and rigorously? 

Reviewer #1: Yes

Reviewer #2: Yes

3. Have the authors made all data underlying the findings in their manuscript fully available?

Reviewer #1: Yes

Reviewer #2: Yes

4. Is the manuscript presented in an intelligible fashion and written in standard English?

Reviewer #1: Yes

Reviewer #2: Yes

5. Review Comments to the Author

Reviewer #1: Overall comment

In this manuscript, the authors analyse the ecological niche of the Orf virus, a cosmopolitan and obligate pathogen of ruminants, in an attempt to discern its limits. I really appreciate the modelling effort and meticulousness applied in this study. However, I have some concerns about a few methodological decisions that might had obscure the results. The failure to identify the niche limits might be related to two issues: i) a slight temporal discrepancy between outbreaks and the environmental data used; and ii) the use of principal components instead of a selection of the original environmental variables. Before acceptance, I strongly encourage the authors to consider these two issues in order to strengthen their conclusions.

Introduction

1st paragraph, line 1: The terms “contagious pustular dermatitis”, “sore mouth” and “contagious ecthyma” are terms referring to the same condition? If it is so, it should be properly stated. For instance, “Contagious pustular dermatitis, sore mouth or contagious ecthyma (CE) are the given names of a viral skin disease…”

1st paragraph, line 3: Please avoid acronyms and abbreviations at the beginning of the sentence (check in the entire text).

1st paragraph, lines 10-12: Divide in two independent sentences. “…on the dorsal aspect of the hands [5]. Infection occurs less…”.

1st paragraph, lines 12-13: Remove colon after “CE infections [4]”. What references the citation [4]? The self-limiting nature? In such a case, the citation should be located with the proper piece of information.

2nd paragraph, lines 8-10: “…from THE environment…”

3rd paragraph: Has a minor contribution to introduce the subject, and can be removed.

Materials and Methods

1st paragraph, lines 3-4: the search terms should not be abbreviated or replaced by acronyms.

2nd paragraph, line 7: “…determination of the centroid of that ADMINISTRATIVE area, calculation of its geographic EXTENSION…”.

3rd paragraph, line 3: Replace [40,39,41] with [39-41].

4th paragraph, lines 1-2: The WorldClim 2.1 database is a climatic consensus comprising the period 1970-2000, whereas your data comprises from 1971-2021. How did the authors deal with this temporal discrepancy?

4th paragraph, lines 5-7: Considering PCA to reduce dimensionality of the environmental variables is an appropriate approach. However, given the results obtained, have the authors consider the possibility of selecting a number of sets of the original variables and repeat their modelling experiments? I know it represents a great deal of work, but the results that could be obtained may reinforce (or not) the conclusions of the present study.

5th paragraph, lines 5-6: Were every combination between the five principal components considered? Or only the set comprising the principal components 1 to 5?

5th paragraph: How were these models assessed?

6th paragraph, lines 1-3: Please explain how these 20 models were selected.

7th paragraph: Please specify the axes (and variables) selected to construct and represent the minimum volume ellipsoids.

Results

1st paragraph, line 2-3: How many candidate models were assessed?

1st paragraph, lines 4-5: Relying the calculation of the AUC on a presence/absence confusion matrix (and thus in true absences), why did the authors use this discrimination metric instead of a more appropriate one, such as the partial ROC? (Peterson et al 2008, doi: 10.1016/j.ecolmodel.2007.11.008).

2nd paragraph, lines 1-4: How many models were finally assessed? And how many were selected in each of the partition experiments?

Figure 4: What does each of the 4 maps indicate? This figure panel should be properly arranged, with each of the maps being individually identified and explained in the figure legend. Furthermore, their interpretation will benefit of superimposing the original occurrence data (3rd paragraph, lines 1-4).

Figure 5: The interpretation of this figure will also benefit of superimposing the original occurrence data (3rd paragraph, lines 4-6).

3rd paragraph, lines 6-11: It is unclear how many models met the omission rate criteria in each of the partition sets. I guess there is one model out of 20 in the three-partition models and none in the two-partition models. Please check and rephrase.

3rd paragraph, lines 11-14: Where are these results presented?

4th paragraph, lines 1-3: Which one of the Maxent models? How was it determined as representative? I dare guess it is irrelevant which one of them. If it is so, you should rephrase as “…used to calibrate Maxent models…”.

Figure 6: This figure panel must be improved, as visualization is less than optimal (especially in the top-left figure). Moreover, axis titles and appropriate references are missing.

4th paragraph: Each of the mentions to Figure 6 should specify to which panel section is referring to.

- It is particularly surprising that, given that the authors have decided to consider the location uncertainty of the outbreaks, the suitability maps provided are not accompanied by the corresponding uncertainty maps. To inform the uncertainty of the modelling procedures is not only strongly recommended, but provides a valuable aid in the interpretation of the results.

Discussion

1st paragraph: Rephrase “In spatial epidemiology, THE lack of knowledge regarding the precise location of outbreaks, and possibilities of hosts, AND moving away from the exact location prior to the initial documentation of infection, can affect the model quality by reducing the effective sample size…”

2nd paragraph, line 3: Remove “statistically”.

3rd paragraph: How can the authors be confident enough to suspect of their models based on those doubtful outbreaks of Orf virus in Russia? If they rely on that data, none of those outbreaks could be included in the modelling procedures?

8th paragraph, line 2: Remove “statistically”.

Reviewer #2: Consider citing 10.1007/s11250-022-03269-6.

Please consider biased reporting of cases especially in developing countries like India, where there are reports from specific locations where there are facilities like veterinary colleges and research institutes.

6. PLOS authors have the option to publish the peer review history of their article (what does this mean?). If published, this will include your full peer review and any attached files.

Reviewer #1: No

Reviewer #2: **Yes: **Renu Gupta

---

## [Author Response · Author response to Decision Letter 0]

25 Jul 2023

Response to Reviewers’ Comments

Reviewer #1 

• In this manuscript, the authors analyse the ecological niche of the Orf virus, a cosmopolitan and obligate pathogen of ruminants, in an attempt to discern its limits. I really appreciate the modelling effort and meticulousness applied in this study. However, I have some concerns about a few methodological decisions that might had obscure the results. The failure to identify the niche limits might be related to two issues: i) a slight temporal discrepancy between outbreaks and the environmental data used; and ii) the use of principal components instead of a selection of the original environmental variables. Before acceptance, I strongly encourage the authors to consider these two issues in order to strengthen their conclusions.

o We really appreciate the reviewer’s encouraging remarks on the manuscript. We would like to express our gratitude to the reviewer for having spent considerable time to review this manuscript. We believe that we have taken adequate care in responding to all of the suggestions by the reviewer… our responses to each of the reviewer’s comments are below.

i) a slight temporal discrepancy between outbreaks and the environmental data used:

We did not want to restrict the records to those that were reported before the year 2000 due to the following reasons:

o Recent increase in reports: We learnt from our literature survey that the factors such as presence of veterinary colleges and hospitals, diagnostic facilities in such institutions, and/or access to farms for veterinarians etc., greatly influence the reporting of Orf prevalence in developing countries. Because of the recent development of infrastructure and diagnostic facilities in developing countries, last decades witnessed reporting of many Orf cases. This is one of the major reasons in the inclusion of many cases from recent decades. 

o Continuous presence in environment and high resilience to environmental degradation; hardy in nature. Orf virus is very resistant to climatic conditions (hardy in nature (Bergqvist et al., 2017)), and can survive in environment for many years. Spyrou and Valiakos (2015) reported that Orf virus can be viable in environment for up to 17 years, and Orf virus has high resilience to environmental degradation (Hsu et al., 2016). These viruses exhibit high ability to survive in harsh environmental conditions (Bukar et al., 2021; Al Saad et al., 2017). Continuous presence of Orf virus in environment was reported to be one of the major reasons behind re-emergence of Orf virus infections (Kumar et al. 2015; Kumar et al 2022).

o Many routes of infection from inanimate objects: Frequent animal movement from one geographical location to another, and capability of Orf virus to remain viable in farm buildings, fomites, farm equipment, and environments for long term (months to years) increases the risk of Orf infections.

Considering all of the above factors, two in particular: continuous presence of Orf virus in environment and high resilience to environmental degradation, we did not want to restrict the occurrence points to those that were reported before the year 2000. Rather, we opted to include as many and as diverse occurrence data as possible, such that our occurrence data were more broadly representative of the environments under which the virus is found. 

 References 

1. Doi: 10.1016/j.vetmic.2015.08.010 

2. Doi: 10.3390/vaccines9111341

3. Doi: 10.9790/2380-1007016469

4. Doi: 10.14737/journal.aavs/2015/3.12.649.676

5. Doi: 10.1007/s11250-022-03269

6. Doi: 10.1093/infdis/jiw307

7. Doi :10.1002/rmv.1932

In the 4th paragraph of the Methods section, we have added a sentence to explain these points and make clear the reasons behind our methodological decisions.

ii) the use of principal components instead of a selection of the original environmental variables

o Our response to this point is explained below in the same reviewer’s comments about our Methods section.

Introduction

• 1st paragraph, line 1: The terms “contagious pustular dermatitis”, “sore mouth” and “contagious ecthyma” are terms referring to the same condition? If it is so, it should be properly stated. For instance, “Contagious pustular dermatitis, sore mouth or contagious ecthyma (CE) are the given names of a viral skin disease…”

o As suggested by the reviewer, the sentence has been modified for greater clarity.

• 1st paragraph, line 3: Please avoid acronyms and abbreviations at the beginning of the sentence (check in the entire text).

o As suggested, ‘CE’ is expanded as “Contagious ecthyma”.

• 1st paragraph, lines 10-12: Divide in two independent sentences. “…on the dorsal aspect of the hands [5]. Infection occurs less…”.

o As suggested, the long sentence is now divided into two independent sentences.

• 1st paragraph, lines 12-13: Remove colon after “CE infections [4]”. What references the citation [4]? The self-limiting nature? In such a case, the citation should be located with the proper piece of information.

o Citation [4] is for self-limiting nature. As suggested, the reference is now properly placed.

• 2nd paragraph, lines 8-10: “…from THE environment…”

o Done.

• 3rd paragraph: Has a minor contribution to introduce the subject, and can be removed.

o As suggested, the 3rd paragraph is deleted.

Materials and Methods

• 1st paragraph, lines 3-4: the search terms should not be abbreviated or replaced by acronyms.

o The search terms (abbreviated and non-abbreviated), mentioned in the materials and methods were used as such for the search, as some of the research is indeed listed under abbreviated names.

• 2nd paragraph, line 7: “…determination of the centroid of that ADMINISTRATIVE area, calculation of its geographic EXTENSION…”.

o The sentence is modified as suggested.

• 3rd paragraph, line 3: Replace [40,39,41] with [39-41].

o The reference format is changed as suggested.

• 4th paragraph, lines 1-2: The WorldClim 2.1 database is a climatic consensus comprising the period 1970-2000, whereas your data comprises from 1971-2021. How did the authors deal with this temporal discrepancy?

o Response to this comment is provided above, under the responses to the reviewer’s general comments. 

• 4th paragraph, lines 5-7: Considering PCA to reduce dimensionality of the environmental variables is an appropriate approach. However, given the results obtained, have the authors consider the possibility of selecting a number of sets of the original variables and repeat their modelling experiments? I know it represents a great deal of work, but the results that could be obtained may reinforce (or not) the conclusions of the present study.

o Thanks for this suggestion. We note that all models do one of two things: they can explain, or they can predict. A model designed to explain is one where we would wish to retain the original variables, as such a model would be maximally interpretable. In this case, however, our goal was to predict suitability across landscapes/regions, and so interpretability of independent (environmental) variables is less necessary. Rather, we would wish to retain a maximum of information in those variables, rather than emphasizing their interpretability. We note that if we could not “predict” using the prediction-optimized PCA variables and prediction-focused modeling approaches, any association with individual (raw) variables is likely to be spurious.

o In this context, we are aiming to use the most climatic information available to us in our analyses. Correlation-based variable removal focuses on pairs of variables correlated at, say r > 0.8 or 0.9… although those pairs of variables mostly covary, they do have some independent variation, and potentially some independent information content, which is lost if we eliminate variables. The PCA, on the other hand, in theory retains that information in new axes that, while less interpretable, retain the full original information. More information about the utility of PCA and our methodological choices has been added to the Methods.

5th paragraph, lines 5-6: Were every combination between the five principal components considered? Or only the set comprising the principal components 1 to 5?

o Here, we were talking about the very early modeling experiment, and in that experiment, we used a set comprising the first five PCs. No combinations were used. We entered into considerably greater depth in subsequent modeling efforts, detailed later in the Methods.

• 5th paragraph: How were these models assessed?

o Method of assessing model performance is mentioned in the methodology section: “Final models were selected from among candidate models based on three criteria: statistical significance, omission rate, and lowest Akaike information criterion (AICc) values, as suggested by Cobos et al. [52]. Statistical significance of models was tested using partial receiver operating characteristic (pROC) statistics [54]. Statistically significant models were filtered using omission rate, retaining only those models for which omission rate was <5% [55]. Statistically significant models with <5% omission rate were ranked based on AICc values and the models that are within 2 AICc units of the minimum were chosen as final models. …….. From the set of final models, the best model was identified as the one with a delta_AICc value of zero (i.e., difference in AIC value between the top model and the remaining models in the final set).”

• 6th paragraph, lines 1-3: Please explain how these 20 models were selected.

o See preceding response. In each of the 20 experiments, the above method was followed. 

• 7th paragraph: Please specify the axes (and variables) selected to construct and represent the minimum volume ellipsoids.

o It is mentioned that first three PCs were used to construct minimum volume ellipsoid. “Seventy percent of replicate occurrence data and first three PCs (PC1, PC2, and PC3) were used to implement the MVE”

Results

• 1st paragraph, line 2-3: How many candidate models were assessed?

o Sixty candidate models were assessed. The sentence has been modified to make this point more clearly.

• 1st paragraph, lines 4-5: Relying the calculation of the AUC on a presence/absence confusion matrix (and thus in true absences), why did the authors use this discrimination metric instead of a more appropriate one, such as the partial ROC? (Peterson et al 2008, doi: 10.1016/j.ecolmodel.2007.11.008).

o Please note that we did not use traditional AUC to assess the model performance, and we used partial ROC tests to choose statistically significant models. It is mentioned in the manuscript: “Statistical significance of models was tested using partial receiver operating characteristic (pROC) statistics” To avoid confusion, the key sentence is modified in the results as follows: “Statistically significant partial AUC ratios (mean ratio 1.116 ± 0.002, all P < 0.001) suggested that the predictions of those models were better than random predictions”

o To further avoid confusion, we have incorporated the following lines in the methodology. “In traditional AUC approach, commission and omission errors are given equal weightage in assessing the model performance [59]. As we used only presence-only data, partial AUC ratio, determined using partial ROC method was employed to analyze the model performance [56]. In this method, x - axis of traditional AUC-ROC (commission error) is replaced with proportion of the whole study area predicted as present to restrict the AUC calculations within the regions where model predicts [56], and under the restricted curve, only the relevant regions that meet the user-defined omission threshold (E = 5%) were considered. Partial AUC ratio is defined as the ratio of AUC of the restricted ROC curve to the AUC of the restricted random expectation curve [56]. Partial AUC ratio values range from 0 to 2, and a value of 1 indicates random performance [56,60]. Partial AUC ratios were averaged across 10 replicates of final models, and represented as mean partial AUC ratios”

• 2nd paragraph, lines 1-4: How many models were finally assessed? And how many were selected in each of the partition experiments?

o A separate supplementary file is prepared (Suppl. file 7), and is cited in the text of the Results; all information related to total number of assessed and selected models were provided. 

• Figure 4: What does each of the 4 maps indicate? This figure panel should be properly arranged, with each of the maps being individually identified and explained in the figure legend. Furthermore, their interpretation will benefit of superimposing the original occurrence data (3rd paragraph, lines 1-4).

o Figure 4 indicates a set of representative models. We are not giving separate description to each of these models as our emphasis is to show consistent and repeated omission of known distributional areas in the predictions. As suggested by the reviewer, occurrence points (white circles) were plotted on each of the models. 

Figure 5: The interpretation of this figure will also benefit of superimposing the original occurrence data (3rd paragraph, lines 4-6).

o As suggested by the reviewer, the occurrence points were plotted on the map.

3rd paragraph, lines 6-11: It is unclear how many models met the omission rate criteria in each of the partition sets. I guess there is one model out of 20 in the three-partition models and none in the two-partition models. Please check and rephrase.

o As suggested by the reviewer, the sentence is modified as follows: “Evaluating the selected models using two partly independent test datasets (IND1 and IND2) sourced from the occurrence pool (i.e., three-partition models) revealed that only one model out of 10 final models met our omission rate criteria (<5%). No model met our omission rate criteria among the 10 models based on two-partition sets. All final models in both two- and three- partition experiments were statistically significant (P < 0.001, Tables 1 and 2)”

3rd paragraph, lines 11-14: Where are these results presented?

o We mentioned a part of our observation in results (3rd paragraph, lines 1-4), and also in the discussion (3rd, 4th, and 5th paragraphs). For clarity, the sentence is modified as follows: “Results of independent final evaluation of selected models thus coincided with manual assessment of predicted models based on distributional knowledge (see discussion for more details): our models failed to predict distributional potential of Orf virus in several areas across the world”

4th paragraph, lines 1-3: Which one of the Maxent models? How was it determined as representative? I dare guess it is irrelevant which one of them. If it is so, you should rephrase as “…used to calibrate Maxent models…”.

o Thanks for this suggestion. As suggested, the sentence is rephrased. 

Figure 6: This figure panel must be improved, as visualization is less than optimal (especially in the top-left figure). Moreover, axis titles and appropriate references are missing.

o As suggested, figure panel is improved and figure legends are modified for better understanding. 

4th paragraph: Each of the mentions to Figure 6 should specify to which panel section is referring to.

o As suggested, figure panels are properly mentioned in the text. 

It is particularly surprising that, given that the authors have decided to consider the location uncertainty of the outbreaks, the suitability maps provided are not accompanied by the corresponding uncertainty maps. To inform the uncertainty of the modelling procedures is not only strongly recommended, but provides a valuable aid in the interpretation of the results.

o As a supplementary figure (8), cited in the text, uncertainty maps are provided. Method of estimating uncertainty is mentioned in the methodology (last sentence of 6th paragraph of methodology), and our observations are mentioned in the results (last sentence of 3rd paragraph of results). 

Discussion

1st paragraph: Rephrase “In spatial epidemiology, THE lack of knowledge regarding the precise location of outbreaks, and possibilities of hosts, AND moving away from the exact location prior to the initial documentation of infection, can affect the model quality by reducing the effective sample size…”

o The sentence has been rephrased.

2nd paragraph, line 3: Remove “statistically”.

o It is removed.

3rd paragraph: How can the authors be confident enough to suspect of their models based on those doubtful outbreaks of Orf virus in Russia? If they rely on that data, none of those outbreaks could be included in the modelling procedures?

o From the given references [71 - 77], it was clear to us that Orf prevalence was noticed in Russia. However, we were not able to get reliable administrative areas in Russia to convert such information into occurrence records by assigning uncertainty. 

8th paragraph, line 2: Remove “statistically”.

o It is removed

Reviewer #2: 

• Consider citing 10.1007/s11250-022-03269-6.

o Thanks for this suggestion. We have cited the paper.

• Please consider biased reporting of cases especially in developing countries like India, where there are reports from specific locations where there are facilities like veterinary colleges and research institutes.

o We really appreciate this comment. We understand the biasedness in reporting in developing countries owing to the availability / non-availability of facilities and infrastructure. As our focus was to develop the global ecological niche of Orf virus, we have used the available records. We have indicated in the last paragraph of the discussion that availability of large sample size may help to improve the niche identification issues related to Orf virus.

---

## [Decision Letter · Decision Letter 1]

16 Aug 2023

PONE-D-23-15637R1An evaluation of the ecological niche of Orf virus (Poxviridae): challenges of distinguishing broad niches from no nichesPLOS ONE

Dear Dr. Nair,

Thank you for submitting your manuscript to PLOS ONE. After careful consideration, we feel that it has merit but does not fully meet PLOS ONE’s publication criteria as it currently stands. Therefore, we invite you to submit a revised version of the manuscript that addresses the points raised during the review process.

We look forward to receiving your revised manuscript.

Kind regards,

Paulo Brandao

Academic Editor

PLOS ONE

Reviewers' comments:

Reviewer's Responses to Questions

**Comments to the Author**

1. If the authors have adequately addressed your comments raised in a previous round of review and you feel that this manuscript is now acceptable for publication, you may indicate that here to bypass the “Comments to the Author” section, enter your conflict of interest statement in the “Confidential to Editor” section, and submit your "Accept" recommendation.

Reviewer #1: (No Response)

Reviewer #3: (No Response)

2. Is the manuscript technically sound, and do the data support the conclusions?

Reviewer #1: Yes

Reviewer #3: (No Response)

3. Has the statistical analysis been performed appropriately and rigorously? 

Reviewer #1: Yes

Reviewer #3: (No Response)

4. Have the authors made all data underlying the findings in their manuscript fully available?

Reviewer #1: Yes

Reviewer #3: (No Response)

5. Is the manuscript presented in an intelligible fashion and written in standard English?

Reviewer #1: Yes

Reviewer #3: (No Response)

6. Review Comments to the Author

Reviewer #1: I am glad to see that most of my comments have been fully considered, and that the manuscript has notably improved. Despite a minor recommendation below, as far as I am concerned, this manuscript is now suitable for its publication.

Concerning the justification for solely using PCA components and not the original environmental variables (previously commented in Materials & Methods, 4th paragraph), and since the issue might be a common concern among readers, I believe this must be included in the Discussion.

Reviewer #3: Article entitled “An Evaluation of the Ecological Niche of Orf Virus (Poxviridae): Challenges of Distinguishing Broad Niches from No Niches.” has some scientific value.

Comments

Main.

To date, there are many scientific works in the field of virology based only on bioinformatics data (in silico). Unfortunately, there is very little confirmation of all bioinformatic data by field studies.

The use of the technique "Peterson, Samy, 2016" needs to be explained and shown to be possible for Poxviridae. The authors (Peterson, Samy, 2016) use it for completely different viruses with completely different ways of spreading in nature.

The authors state "thus enhances our ability to deal with varying degrees of uncertainty associated with precise localization." Is there at least one confirmation of this postulate in the scientific literature? If so, why is it not cited? Or is it just unconfirmed thoughts of the authors?

In the study, the authors use the "NicheA" platform used for niche visualization in environmental space for a specific virus - Orf Virus. To confirm the capabilities of this technique, several literary sources are used (numbers 50-54). However, these articles do not use viruses as a model; moreover, they were published several years earlier than the proposed methodology. How can the proposed methodology be justified in this way?

Despite the unsuccessful predictions of previous models, the authors argue that there is an opportunity to improve the ability to predict epidemics, but their description and comparison requires more meaningful evaluation data.

Minor points

The NicheA program is practically not used for analysis in virology, therefore, additional explanations for the choice of this software are required.

How correct is it to cite an article (Low et al 2020) in the legend for Figure 5?

The authors compare different methods of virus ecology analysis. For a better perception, such an analysis should be accompanied by a table with comparisons of various programs.

7. PLOS authors have the option to publish the peer review history of their article (what does this mean?). If published, this will include your full peer review and any attached files.

Reviewer #1: **Yes: **Pablo F. Cuervo (Departamento de Parasitología, Universidad de Valencia)

Reviewer #3: No

---

## [Author Response · Author response to Decision Letter 1]

19 Sep 2023

Response to Reviewers’ Comments

Reviewer #1 

• I am glad to see that most of my comments have been fully considered, and that the manuscript has notably improved. Despite a minor recommendation below, as far as I am concerned, this manuscript is now suitable for its publication.

o We express our deep gratitude to the reviewer for helping us to improve the manuscript considerably. 

• Concerning the justification for solely using PCA components and not the original environmental variables (previously commented in Materials & Methods, 4th paragraph), and since the issue might be a common concern among readers, I believe this must be included in the Discussion.

o Thanks for this suggestion. As suggested by the reviewer, we incorporated the details related to PCA in the first paragraph of the Discussion.

Reviewer #3: 

• To date, there are many scientific works in the field of virology based only on bioinformatics data (in silico). Unfortunately, there is very little confirmation of all bioinformatic data by field studies.

o We appreciate the reviewer’s statement, but we are not entirely sure of their point. In our paper, we are trying to take results from detailed laboratory studies, and interpret them in terms of the suitability of landscapes/regions for Orf virus across the world by using an ecological niche modeling approach. 

o In the abstract, we have mentioned clearly the challenges involved in interpreting our results as follows: “These results suggest two possibilities: that the niche models fail to identify niche limits that constrain the virus, or that the virus has no detectable niche, as it can be found throughout the geographic distributions of its hosts.”

o In the manuscript, we suggested the importance of cross-verifying the model predictions with existing field-based distributional knowledge in several places. In the Results: “However, manual assessment of models (Fig 4) based on the available distributional knowledge of Orf virus revealed that all of our models failed consistently to predict known occurrences in many parts of the world…” and “Results of independent final evaluation of selected models thus coincided with manual assessment of predicted models based on distributional knowledge (see Discussion for more details): our models failed to predict distributional potential of Orf virus in several areas across the world.”

o In the Discussion is this text: “We sought to answer the niche question for Orf virus via two assessments: (1) Do generated models show non-random patterns of model failure? (2) Can we see consistent and repeated patterns of omission in the models generated? In this latter step, we assessed the models developed by integrating additional distributional knowledge, and noticed that significant parts of the known distribution of Orf virus were not represented in the geographic space visualizations.”

o More generally, we point to the title of our manuscript, indicating that there are challenges associated with identifying broad niches from no niches in the case of globally distributed pathogens. We believe that this study will help researchers to acknowledge certain potential caveats in interpreting model results realistically. In this sense, we believe that we are in agreement with the reviewer’s comment. 

• The use of the technique "Peterson, Samy, 2016" needs to be explained and shown to be possible for Poxviridae. The authors (Peterson, Samy, 2016) use it for completely different viruses with completely different ways of spreading in nature.

o Peterson and Samy (2016) used buffer zones of varying sizes around each occurrence record to address uncertainty in model outputs. The method employed is not species-specific, which is to say that, if uncertainty exists in the spatial localization of species occurrence records, this method helps researchers to incorporate the uncertainty directly into the analysis process. This method was particularly suitable in our case as we did not have very precise geographical coordinates for many records. 

• The authors state "thus enhances our ability to deal with varying degrees of uncertainty associated with precise localization." Is there at least one confirmation of this postulate in the scientific literature? If so, why is it not cited? Or is it just unconfirmed thoughts of the authors?

o The preceding part of the sentence had the reference that the reviewer is seeking. However, to avoid confusion, we have now placed this reference at the end of the sentence.

• In the study, the authors use the "NicheA" platform used for niche visualization in environmental space for a specific virus - Orf Virus. To confirm the capabilities of this technique, several literary sources are used (numbers 50-54). However, these articles do not use viruses as a model; moreover, they were published several years earlier than the proposed methodology. How can the proposed methodology be justified in this way?

o Sorry for the confusion. The references to which the reviewer refers were not used for supporting the capabilities of NicheA. Rather, those references were key in supporting underlying assumption, such as the convex nature of species’ fundamental niches, which is the key idea in the NicheA algorithm. 

o Considering the reviewer’s suggestion to incorporate more works related to niche modeling studies of viruses in which NicheA was employed, we have included additional references (117, 118 & 119) in line 3 of paragraph 7 in discussion.

• Despite the unsuccessful predictions of previous models, the authors argue that there is an opportunity to improve the ability to predict epidemics, but their description and comparison requires more meaningful evaluation data.

o We have mentioned in the article that the ENM approaches have limitations in developing realistic models of globally distributed viruses: “These results suggest two possibilities: that the niche models fail to identify niche limits that constrain the virus, or that the virus has no detectable niche, as it can be found throughout the geographic distributions of its hosts.” (see abstract, lines 18-20)

o Considering the potential limitations and inferior quality of models developed, we concluded in the discussion that “However, confirming whether or not Orf virus has a detectable niche separate from that of its host remains a challenge, awaiting much larger sample sizes, and information on host-breadth [35] (i.e., accessible host populations for Orf virus to infect and survive in the absence of dispersal barriers). Host-breadth identification may reveal relevant biotic factors that can inform component-based ENM of pathogens [35].

o As such, in this article, our focus was to assess the models’ capability to predict, and we noted that models failed to predict suitability in several known-distributional areas across the world. We believe that this study will help the researchers to acknowledge certain potential caveats in realistically interpreting model results. As such, we believe that our Discussion of our results is actually in agreement with the reviewer’s comments.

• The NicheA program is practically not used for analysis in virology, therefore, additional explanations for the choice of this software are required.

o NicheA has indeed been used for assessing the ecological niches of viruses in environmental space, previously, as can be appreciated from the citations listed above. More broadly, though, NicheA is just a tool for visualization and analysis of data, and is not the focus of this paper. In this paper, we marshall several novel tools for this analysis … but the point of the paper is to assess whether a detectable ecological niche exists for a globally distributed virus.

• How correct is it to cite an article (Low et al 2020) in the legend for Figure 5?

o We do not think that it is uncommon or inappropriate to cite references in the figure legends, but the editors of the journal can decide whether that is something to be avoided.

• The authors compare different methods of virus ecology analysis. For a better perception, such an analysis should be accompanied by a table with comparisons of various programs.

o We have added such a table, in the supplementary materials.

---

## [Editor Report · Decision Letter 2]

10 Oct 2023

An evaluation of the ecological niche of Orf virus (Poxviridae): challenges of distinguishing broad niches from no niches

PONE-D-23-15637R2

Dear Dr. Nair,

We’re pleased to inform you that your manuscript has been judged scientifically suitable for publication and will be formally accepted for publication once it meets all outstanding technical requirements.

Kind regards,

Paulo Brandao

Academic Editor

PLOS ONE

---

## [Editor Report · Acceptance letter]

17 Oct 2023

PONE-D-23-15637R2 

An evaluation of the ecological niche of Orf virus (*Poxviridae*): challenges of distinguishing broad niches from no niches 

Dear Dr. Nair:

I'm pleased to inform you that your manuscript has been deemed suitable for publication in PLOS ONE. Congratulations! Your manuscript is now with our production department. 

Kind regards, 

on behalf of

Dr. Paulo Brandao 

Academic Editor

PLOS ONE